# Ursolic Acid Formulations Effectively Induce Apoptosis and Limit Inflammation in the Psoriasis Models In Vitro

**DOI:** 10.3390/biomedicines12040732

**Published:** 2024-03-25

**Authors:** Ewa Bielecka, Natalia Zubrzycka, Karolina Marzec, Anna Maksylewicz, Maja Sochalska, Agnieszka Kulawik-Pióro, Elwira Lasoń, Karolina Śliwa, Magdalena Malinowska, Elżbieta Sikora, Krzysztof Nowak, Małgorzata Miastkowska, Tomasz Kantyka

**Affiliations:** 1Malopolska Centre of Biotechnology, Jagiellonian University, Gronostajowa 7A, 30-387 Cracow, Poland; ewa.bielecka@uj.edu.pl (E.B.); natalia.zubrzycka@doctoral.uj.edu.pl (N.Z.); karolinagabriela.marzec@student.uj.edu.pl (K.M.); anna.maksylewicz@uj.edu.pl (A.M.); 2Faculty of Biochemistry, Biophysics and Biotechnology, Jagiellonian University, Gronostajowa 7, 30-387 Cracow, Poland; maja.sochalska@uj.edu.pl; 3Doctoral School of Exact and Natural Sciences, Jagiellonian University, Gronostajowa 7, 30-387 Cracow, Poland; 4Department of Organic Chemistry and Technology, Faculty of Chemical Engineering and Technology, Cracow University of Technology, Warszawska 24, 31-155 Cracow, Poland; agnieszka.kulawik-pioro@pk.edu.pl (A.K.-P.); elwira.lason@pk.edu.pl (E.L.); karolina.sliwa@pk.edu.pl (K.Ś.); magdalena.malinowska@pk.edu.pl (M.M.); esikora@pk.edu.pl (E.S.); malgorzata.miastkowska@pk.edu.pl (M.M.); 5Wellnanopharm, Jerzego Samuela Bandtkego 19, 30-129 Cracow, Poland; nowak.krzysztof@autograf.pl

**Keywords:** psoriasis, keratinocytes, hyperproliferation, inflammation, ursolic acid, topical formulation

## Abstract

Psoriasis, a prevalent inflammatory skin disorder affecting a significant percentage of the global population, poses challenges in its management, necessitating the exploration of novel cost-effective and widely accessible therapeutic options. This study investigates the potential of ursolic acid (UA), a triterpenoid known for its anti-inflammatory and pro-apoptotic properties, in addressing psoriasis-related inflammation and keratinocyte hyperproliferation. The research involved in vitro models employing skin and immune cells to assess the effects of UA on psoriasis-associated inflammation. The presented research demonstrates the limiting effects of UA on IL-6 and IL-8 production in response to the inflammatory stimuli and limiting effects on the expression of psoriatic biomarkers *S100A7*, *S100A8*, and *S100A9*. Further, the study reveals promising outcomes, demonstrating UA’s ability to mitigate inflammatory responses and hyperproliferation of keratinocytes by the induction of non-inflammatory apoptosis, as well as a lack of the negative influence on other cell types, including immune cells. Considering the limitations of UA’s poor solubility, hybrid systems were designed to enhance its bioavailability and developed as hybrid nano-emulsion and bi-gel topical systems to enhance bioavailability and effectiveness of UA. One of them in particular–bi-gel–demonstrated high effectiveness in limiting the pathological response of keratinocytes to pro-psoriatic stimulation; this was even more prominent than with ursolic acid alone. Our results indicate that topical formulations of ursolic acid exhibit desirable anti-inflammatory activity in vitro and may be further employed for topical psoriasis treatment.

## 1. Introduction

Psoriasis is a common inflammatory skin disease that affects nearly 2–5% of the population worldwide. Epidemiological data on psoriasis prevalence pinpoint a slightly higher occurrence rate in the Caucasian population, especially in people living in countries more distant from the equator [1]. Clinical symptoms of psoriasis are characterized by an acute inflammatory reaction in the first phase, which turns into a chronic form. Psoriasis is characterized by three typical histological features: (i) epithelial hyperplasia caused by keratinocyte hyperproliferation, (ii) dilated, enhanced blood vessel formation in the dermis, and (iii) accumulation of inflammatory cells, mainly in the dermis. Due to the chronic, systemic inflammatory state, psoriatic patients are more prone to develop other conditions like psoriatic arthritis, hypertension, diabetes, and cardiovascular diseases including atherosclerosis and stroke [2]. The current hypothesis regarding psoriasis development indicates primary activation of the innate immune system, based on the impaired response of keratinocytes, neutrophils, and dendritic cells. Damaged and overstimulated keratinocytes are the source of chemotactic cytokines, which recruit the PMNs to the site of inflammation. It has been shown that high levels of neutrophils in psoriatic lesions are the source of free DNA released via NETosis, which forms the complexes with LL-37 antimicrobial peptide, leading to strong stimulation of plasmacytoid dendritic cells (pDCs). The disturbance in the functioning of the adaptive immune system, including excessive activation of dendritic cells and, in turn, Th_1_/Th_17_ lymphocytes, results in the overproduction of cytokines, especially IL-17A, IL-22, TNF-α, and interferon γ, which generate the full clinical outcome of the disease [3,4,5].

Taking into account the engagement of both innate and adaptive immunity in development of psoriasis, several experimental setups are in use. The two most common are utilization of (i) a cytokine cocktail characteristic of psoriasis—IL-17A, IL-22, IL-1α, oncostatin M, and TNFα—so called M5 [6,7], or (ii) imiquimod [8]. The M5 mix is mainly used in studies engaging keratinocytes as effector cells in psoriasis, as it induces a psoriasis-like phenotype of those cells, including hyperproliferation, upregulation of cytokines and chemokines, and antimicrobial peptide production with simultaneous downregulation of keratinocytes differentiation markers.

Current anti-psoriasis treatments focus on reducing the inflammation state and the limitation of keratinocyte hyperproliferation. Therapies include metotrexat (cytostatic), cyclosporin (immunosuppressant), and highly effective biological therapeutics like monoclonal antibodies against TNF-α, IL-17, and IL-23 [9]. Still, due to the adverse effects of the mentioned drugs and the very high cost of monoclonal antibody-based treatment, searching for new treatment options is necessary. Delivery of cheap and widely available formulations characterized by moisturizing, exfoliating, anti-inflammatory, and anti-proliferation properties is desired. For that reason, many naturally derived plant compounds are applied in psoriasis treatment. Ursolic acid (3β-hydroxyurs-12-en-28-oic acid; UA) is a pentacyclic triterpenoid, a secondary metabolite identified in many commonly used plants, including herbs (thyme, rosemary, lavender, oregano, and mint), and mainly in fruit peels. Ursolic acid can mediate anti-inflammatory, anti-oxidant, anti-obesity, and anti-carcinogenic effects in in vitro models and animal in vivo models [10]. The anti-cancer activity of ursolic acid is related to the inhibition of proliferation and induction of apoptosis of cancer cells [11,12,13]. Triterpenoid acids present in *Melissa officinalis* extracts, including ursolic acid, 2α-hydroxy-ursolic acid, pomolic acid, and oleanolic acid, have been shown to be responsible for anti-psoriatic effects in a psoriatic mice model, significantly inhibiting epidermal hyperplasia and scaling [14].

Ursolic acid is characterized by very poor solubility in both hydrophilic and hydrophobic solvents, resulting in poor absorption, very low bioavailability, and thus limited biological activity [15]. The toxicity of the compound is apparently low, as even prolonged oral administration does not show adverse effects. Taking into account anti-inflammatory and anti-proliferative properties and the preferable safety profile of ursolic acid [16], it is a good candidate for a topical formulation. Due to triterpenoid acid’s poor water solubility, the incorporation of UA into alternate drug delivery systems has been explored. Compositions based on colloid nanocarriers, including polymeric nanoparticles, liposomes, polymeric micelles, and nanostructured lipid carriers, were found to be efficient at improving its solubility [17].

Considering the benefits of the above-mentioned colloidal nanocarriers and their ability to encapsulate hydrophobic active substances, the object of the authors’ previous research was to design and obtain novel, hybrid systems as vehicles for ursolic acid. The physicochemical properties of the designed formulations were characterized in our previous research and their safety was confirmed by in vitro assays [18].

The aim of the presented work was to investigate the beneficial effects of ursolic acid on psoriasis-related inflammation and keratinocyte proliferation using in vitro models based on the skin and immune cells. The findings were implemented and allowed the creation of complex topical formulations with improved bioavailability and effectiveness in psoriasis in vitro models.

## 2. Materials and Methods

### 2.1. Preparation of Formulations

To obtain the hybrid systems containing ursolic acid (Merck, Poznań, Poland), a multistage procedure was applied. First, the nano-emulgel (NG) was prepared by incorporating the nano-emulsion (NE) into the hydrogel matrix (H). Next, the macroemulsions serum (S), cream (C), balm (B), and oleogel (O) were developed. Finally, the hybrid systems were obtained by combining the base formulation (cream, serum, balm, and oleogel) with nano-emulgel in various weight ratios (from 10:90 to 90:10 wt.%). In the case of all formulations, the concentration of ursolic acid was 0.5% wt.%. The obtained topical formulations were characterized by conducting morphological, rheological, texture, and stability analyses. To determine the safety and effectiveness of the prepared ursolic acid carriers, in vitro studies on human keratinocyte cell-like HaCaT cells were performed with cytotoxicity analysis. Moreover, kinetic studies of ursolic acid release from the obtained systems were conducted. Preparations selected for further research based on their physicochemical analysis are presented in Table 1.

All details regarding composition, methods of preparation, and characterization of all obtained formulations were presented in our previous studies [18]. The general procedure is presented on Figure 1.

### 2.2. Cell Culture

HaCaT cells were cultured in DMEM medium (Life Technologies, Warsaw, Poland) containing 10% FBS (Life Technologies, Warsaw, Poland), 100 U/mL penicillin (Life Technologies, Warsaw, Poland) and 100 ug/mL streptomycin (Life Technologies, Warsaw, Poland) in 37 °C/5% CO_2_. Prior to each experiment, cells were seeded on plates in DMEM medium without any supplementation.

### 2.3. Effect of Ursolic Acid and Formulation on Cytokine Release of HaCaT upon Stimulation with M5 Cytokine Mix

HaCaT cells were seeded on 48-well plates (100,000 cells/well) a day prior to the experiment and stimulated with different concentrations of ursolic acid formulations in the presence of an M5 cytokin mix consisting of 1 ng/mL of each IL-17A, IL-22, oncostatin M, IL-1α, and TNF-α (BioLegend, BioCourse.pl, Katowice, Poland) diluted in DMEM medium. After 24 h, cell media were collected and stored at −20 °C for further analysis.

### 2.4. RNA Isolation and Quantitative Polymerase Chain Reaction (qPCR)

HaCaT cells were seeded on 24-well plates (2 × 10^5^ cells/well) a day prior to the experiment. Cells were incubated with various concentrations of ursolic acid in the presence of 1 ng/mL M5 cytokine mix diluted in DMEM medium supplemented with 1% FBS. After 24 h, the medium was removed, cells were washed once with PBS, and total RNA was extracted using an ExtractMe Total RNA isolation kit (Blirt, Gdańsk, Poland). mRNA concentration was quantified using a NanoDrop 2000 UV-Vis spectrophotometer. RNA was reverse transcribed using a High-Capacity cDNA Reverse Transcription Kit (Applied Biosystems, Life Technologies, Warsaw, Poland). PowerUp SYBR Green PCR mix (Applied Biosystems, Life Technologies, Warsaw, Poland ) was used for qPCR reactions that were performed on a CFX96 Touch™ Real-Time PCR Detection System (Bio-Rad, Warsaw, Poland) and the obtained results were analyzed using the CFX Manager (Bio-Rad, Warsaw, Poland). mRNA expression relative to housekeeping genes (*EEF2*) was calculated using the ΔΔCT method. Sequences of the primers used are listed in Table 2.

### 2.5. Proliferation of HaCaT Cell Line upon Stimulation with M5 Cytokin Mix in Presence of Ursolic Acid Formulations

HaCaT cells were seeded on 96-well plates (5000 cells/well) a day prior to the experiment. Cells were incubated with various concentrations of ursolic acid formulations in the presence of 1 ng/mL M5 cytokine mix diluted in DMEM medium supplemented with 1% FBS. After 72 h, the medium was removed, and cells were washed once with PBS and incubated with 200 µL/well of 0.5 mg/mL Thiazolyl Blue Tetrazolium Bromide (Merck, Poznań, Poland) in DMEM for up to 20 min at 37 °C/5% CO_2_. The medium was removed again, and formazan crystals were dissolved in 120 µL of isopropanol acidified with 5 mM hydrochloric acid (Avantor Performance Materials, Gliwice, Poland). Quantities of 90 µL of samples from each well were transferred to a new, transparent 96-well plate. Absorbance of samples was measured using Spectra Max Gemini EM (Molecular Devices, San Jose, CA, USA) at 570 nm. Results were calculated as a percentage of the untreated control (cells in DMEM medium).

### 2.6. ELISA

Levels of pro-inflammatory cytokines IL-6 and IL-8 in the cell culture media were measured using commercial ELISA kits (R&D, Bio-Techne, Warsaw, Poland) according to the manufacturer’s instructions. Half-area, high-binding, 96-well plates (Greiner, Merck, Poznań, Poland) were coated o/n with appropriate coating antibodies diluted in PBS. On the next day, wells were washed with washing buffer (0.05% Tween in PBS) and blocked for 1 h at RT with 150 µL of 1% BSA diluted in PBS. Wells were washed and 50 µL of samples was added to the wells and incubated for 2 h in RT. After a thorough wash, wells were incubated with the primary antibodies conjugated with biotin for 2 h and washed again. Then, the wells were incubated with streptavidin conjugated with HRP for 30 min. Prior to developing, wells were washed five times, and 50 µL of TMB substrate solution (TMB Substrate Reagent Set, BD, Warsaw, Poland) was added per well. After the signal was developed sufficiently, the reaction was stopped with 25 µL 2 N H_2_SO_4_ (Avantor Performance Materials, Gliwice, Poland). The absorbance was measured at 450 nm and 570 nm.

### 2.7. Human Neutrophil Isolation

Peripheral human blood from healthy donors for PMN isolation was obtained from Red Cross, Krakow, Poland. The Red Cross de-identified blood materials as appropriate for the confidentiality assurance of human subjects. Donors provided written informed consent for the collection of samples and subsequent cell isolation and analysis. Thus, this study adheres to appropriate exclusions from the approval of human subjects, according to current regulations. Neutrophils were isolated from fractions enriched with granulocytes, which were harvested by centrifugation over a density gradient using a lymphocyte separation medium as described by Bryzek et al. 2019 [19]. Briefly, neutrophils and erythrocytes were collected as the high-density fraction and separated after 30 min of incubation with 1% polyvinyl alcohol (Avantor Performance Materials, Gliwice, Poland ). Neutrophils were collected from the upper layer and, after centrifugation (280× *g*, 10 min, RT), the residual erythrocytes were removed by lysis in water. Neutrophils were resuspended in 2% FBS RPMI 1640 medium (Life Technologies, Warsaw, Poland) without antibiotics.

### 2.8. Stimulation of Human Neutrophils with Ursolic Acid and M5

Human neutrophils were seeded at 1 × 10^6^ cells per 1 mL. The neutrophils were then stimulated with different concentrations of ursolic acid in the presence of a M5 cytokine mixture consisting of 1 ng/mL each of IL-17A, IL-22, oncostatin M, IL-1α, and TNF-α diluted in RPMI 1640 medium. After 24 h, the cell medium was collected and stored at −20 °C for further analysis, and cells were harvested for flow cytometry analysis.

### 2.9. Annexin V and Propidium Iodide Staining

The percentage of viable cells was determined using Annexin V-APC and/or propidium iodide (BioLegend, BioCourse.pl, Katowice, Poland) according to the manufacturer’s protocol, followed by flow cytometry analysis on the BD LSRFortessa^TM^, (Becton Dickinson, Warsaw, Poland) as described previously [20]. The flow cytometry data were analyzed with FlowJo Software (Version X for Windows, FloJo LLC, Ashland, OR, USA).

### 2.10. Reactive Oxygen Species Production

Reactive oxygen species (ROS) production was determined as described previously [20]. Briefly, cells were harvested and 10^5^ cells were stained by incubation with 20 µM 2′,7′-dichlorofluorescein diacetate (DCF) (Merck, Poznań, Poland) for 30 min at 37 °C and washed with ice-cold PBS (Sigma-Aldrich). Fluorescence was analyzed by flow cytometry on the BD LSRFortessa^TM^. The FACS data were analyzed with FlowJo Software (Version X for Windows, FloJo LLC).

### 2.11. Statistical Analysis

The results were analyzed with GraphPad Prism software (v. 10.2.1) (Boston, MA, USA) and are presented as the mean of each experiment ± SEM. Statistical significance was evaluated with the built-in one-way ANOVA test. Results were considered statistically significant if *p*-values < 0.05. *—*p* = 0.05–0.011; **—*p* <0.01; ***—*p* < 0.001; ****—*p* < 0.0001.

## 3. Results

The keratinocyte-like HaCaT cell model, stimulated with the M5 mixture of pro-psoriatic cytokines (17A, IL-22, oncostatin-M, IL-1α, and TNF-α; 1 ng/mL each) was applied to determine the effect of the ursolic acid (UA) on the pro-inflammatory signaling under the psoriatic conditions. The stimulation of the HaCaT cells was observed, illustrated by the increase in IL-6 and IL-8 release to the culture medium after 24 h of stimulation. The presence of ursolic acid limited the observed pro-inflammatory HaCaT response, as the increasing concentrations of UA resulted in the decrease in the observed levels of both IL-6 and IL-8, reaching a reduction of nearly 50% for 2.5 µM and 5 µM, respectively, when compared to the cells stimulated with M5 mix only (Figure 1). Importantly, no cytotoxic effects of ursolic acid up to a 10 µM concentration in the fully confluent HaCaT culture were observed, indicating that the decrease in the production of pro-inflammatory cytokines is not related to the decrease in cell viability [18].

To further evaluate ursolic acid effects on the psoriasis-like phenotype of the HaCaT cells, the gene expression of epithelial psoriasis markers was analyzed and varying effects of ursolic acid were observed (Figure 2). Namely, psoriasin expression (*S100A7*) was significantly induced by stimulation of HaCaT culture with the M5 mix. Increasing concentrations of ursolic acid showed a tendency to decrease this stimulation, reaching the level of statistical significance at a 10 µM UA concentration (Figure 2A). Similarly, the expression of calgranulin A and B (*S100A8* and *S100A9*) was induced by M5 in HaCaT cell culture, and this induction was limited in the presence of ursolic acid. The effect was more pronounced in comparison to psoriasin, as 1 µM UA reduced expression of S100A8 by nearly 2-fold, while 2.5 µM UA reduced *S100A9* expression by 25% and 5 µM UA led to a 50% reduction in calgranulin B expression (Figure 2C,D). In contrast, expression of *CXCL1* was stimulated by M5, albeit moderately (a 4-fold increase in comparison to the unstimulated cells), but effects of UA were not observed regardless of the UA concentration. Even more strikingly, expression of *CCL20* was stimulated by M5 and further augmented by the increasing concentrations of ursolic acid, reaching a nearly 2-fold induction at a 1 µM UA concentration, when compared to M5 mix alone (Figure 2B).

This indicates that effects of ursolic acid limit the inflammatory state, targeting the production and release of pro-inflammatory cytokines, and may limit the neutrophile recruitment by reduction in calgranulin A and B expression. At the same time, however, these effects may enhance production of CCL20 and increase the potential to recruit lymphocytes to the region of psoriatic lesions.

One of the hallmarks of psoriasis is the excessive proliferation of epithelial cells. Therefore, the effect of ursolic acid on the HaCaT cell proliferation was investigated. Again, the M5 stimulation model was employed in the low-confluency HaCaT cultures. Indeed, a significant induction of HaCaT proliferation was observed, as the M5 cytokine mixture induced a 30% increase in cell numbers when compared to the untreated control. Importantly, in the presence of the ursolic acid this effect was reduced and, at 0.5 µM ursolic acid, a tendency to limit M5-induced HaCaT proliferation was observed. At 2.5 µM UA, cell proliferation was restored to the control level and, most significantly, no cytotoxic effects on the HaCaT population up to 5 µM ursolic acid were observed, even in these highly proliferating, low-confluence HaCaT cultures (Figure 3).

The reduction in the cell viability at higher concentrations of ursolic acid (10–20 µM) was noticeable. To describe the effect of these high concentrations of ursolic acid on HaCaT cells and to determine the characteristics of cell death, flow cytometry analysis was employed, with propidium iodide and Annexin V double-stained cells, allowing the determination of necrosis and apoptosis ratios. After 24 h of stimulation of confluent HaCaT culture with increasing concentrations of UA (0–40 µM), the significant induction of cell death was observed. Lower concentrations (5 and 10 µM) of ursolic acid were tolerated. At higher concentrations, however, a significant induction of cell mortality was observed, as remarkably at 20 µM UA, only ~50% of cells remained alive (Figure 4).

This change in cell survival was attributed mainly to the increase in apoptotic cells, as a fraction of Annexin V-positive cells increased to >30% in the presence of 20 µM ursolic acid. A further increase in UA concentrations enhanced the observed effects, as indicated by the increase in the fraction of either Annexin V-positive, or double-positive Annexin V^+^/PI^+^ cells, corresponding to early and late apoptotic cells, respectively. Importantly, even for the highest, 40 µM, ursolic acid concentration, necrotic cells were a minor fraction of all cells, and did not exceed 14% of the whole cell population, indicating the predominantly apoptotic characteristic of cell death induced by ursolic acid in the HaCaT culture.

Neutrophils are critical effector cells in inflammatory reaction in general, and in psoriatic response in particular. Therefore, the effects of ursolic acid on the functional aspects of human neutrophil cells were investigated. Firstly, the effect of ursolic acid on the release of pro-inflammatory mediators by human neutrophils was tested. An ELISA assay was used to estimate the effect of ursolic acid on human neutrophil cytokine production, both in PMNs stimulated with ursolic acid alone and with an M5 cytokine mix and UA. No significant changes in the production of IL-17, IL-22, INF-β, and TNF-α were observed, as these cytokines were not detectable in our assays. However, stimulation of IL-8 production was noticed when neutrophils were stimulated with ursolic acid in a concentration-dependent manner. Observed culture medium levels of IL-8 increased, with 300 pg/mL for 2.5 µM UA, and reaching a remarkable level of 700 pg/mL for 5 µM ursolic acid. At 10 µM UA, a significant drop in the released IL-8 concentrations was observed, as the measured levels were approximately 50 pg/mL and were comparable to those of the unstimulated control (Figure 5A). Further, a synergy of ursolic acid and M5 stimulation was recognized. Neutrophils stimulated with M5 alone increased production of IL-8, but to the moderate levels of approximately 100 pg/mL. When co-stimulated with increasing concentrations of ursolic acid, we observed somewhat sharp increases in IL-8 production, reaching 300 pg/mL for 1 µM, and 1500 pg/mL for 2.5 µM and 5 µM ursolic acid. Again, a steep decrease was noticed when M5-stimulated neutrophils were incubated with 10 µM ursolic acid, as observed levels of IL-8 did not exceed 500 pg/mL (Figure 5B).

Startled by this dramatic breakdown of the ursolic acid concentration-dependent increase in IL-8 production, one could hypothesize that this effect may be attributed to the increased mortality of PMN cells at high (10 µM) ursolic acid concentrations. Therefore, apoptotic cell death of human neutrophils, stimulated with either ursolic acid alone or in combination with M5 stimulation, was analyzed using Annexin V staining and flow cytometry. Increasing concentrations of ursolic acid decreased the fraction of Annexin V-negative cell counts at 5 and 10 µM concentrations. Only 25% of the cell population remained Annexin V-negative at a 10 µM concentration of ursolic acid, demonstrating an increase in apoptotic cells in the population (Figure 6A,C). A similar effect was observed upon concurrent stimulation with M5 and UA. Stimulation with a cytokine mix decreased the fraction of Annexin V-positive cells, indicating the increased neutrophil survival expected under inflammatory conditions. At the same time, incubation with increasing concentrations of ursolic acid induced apoptotic cell death, observed as a dramatic decrease in the Annexin V- negative cell population. Under these pro-inflammatory conditions, 2.5 µM UA seemed to restore the number of apoptotic cells to that observed in the population not stimulated by the M5 cytokine mix. At 5 µM ursolic acid, the fraction of non-apoptotic, Annexin V-negative cells was reduced to 40% of the whole cell population, and fewer than 20% of all PMN cells remained Annexin V-negative at 10 µM UA (Figure 6B,D), indicating the proapoptotic effects of ursolic acid in both normal and inflammatory environments. The observed pro-apoptotic effects of ursolic acid corelated with the drop in cytokine production, suggesting that an increase in apoptotic cell death of human neutrophils was a likely culprit. This further confirms the anti-inflammatory characteristic of the cell death induced by UA. Of note, this effect is similar to the stimulation of apoptotic cell death in keratinocyte-like HaCaT cells, as described above; however, it was observed at significantly lower ursolic acid concentrations (5 µM vs. 20 µM), indicating potential selectivity of this proapoptotic effect towards inflammatory neutrophils.

To follow up the neutrophil function analysis, the impact of ursolic acid stimulation on the neutrophil oxygen burst was investigated. To this end, flow cytometric analysis of DCF-labeled PMN culture was used. This cell-permeating dye interacts with reactive oxygen species (ROS), allowing identification of the cell population characterized by the increased production of ROS. Again, two parallel models were investigated: stimulation of neutrophils with graded doses of ursolic acid alone or in combination with a pro-inflammatory cytokine mixture M5. Indeed, the elevation in the DCF-positive fraction of cells upon incubation with ursolic acid was observed. This markup was concentration dependent, as a 2.5 µM UA concentration showed the trend of increasing DCF^+^ cells, and this effect was statistically significant for 5 µM and 10 µM ursolic acid, where approximately 15% and 20% of the cell population, respectively, was DCF- positive, revealing a substantial increase in ROS production upon ursolic acid treatment (Figure 7A,C).

A similar effect was noted for cells stimulated with UA and an M5 mix together; however, we did not observe an increase in DCF-positive cells in the presence of M5 alone. Furthermore, concurrent treatment with ursolic acid and M5 did not reveal any synergy, which corresponded well with data from the UA treatment alone, reaching almost identical levels of approximately 20% DCF-positive cells for 10 µM UA in the presence of the M5 mix. This indicates that pro-inflammatory stimulation of PMN cells with the M5 mixture does not elevate ROS production, and ursolic acid has the potential to increase ROS generation in human neutrophils, albeit moderately (Figure 7B,D).

As indicated above, an in vitro model of pro-inflammatory stimulation using the M5 mix was validated and resulted in the induction of the psoriatic-like phenotype in the HaCaT keratinocyte-like cell line. An increase in the production of pro-inflammatory cytokines, upregulation of major psoriatic markers, and hyperproliferation of cells were observed, consistent with the changes observed in psoriasis. Application of UA resulted in the resolution of the psoriatic features induced by M5 in the stimulated HaCaT cell line.

Several formulations of ursolic acid intended to be used as topical support treatment for psoriatic patients have been recently developed in our research. These compositions, based on hybrid delivery systems, nano-emulgel–macroemulsion and nano-emulgel–oleogel (bi-gel), as the vehicles for ursolic acid, were proven to be well tolerated in the in vitro cell culture models and in vivo skin irritation tests. Herein, their biological activity as anti-inflammatory delivery systems of ursolic acid was validated. To this end, all prepared formulations were tested in an M5 cytokine mix stimulation model, where HaCaT cells were incubated with a psoriasis-related cytokine cocktail, followed by treatment with the respective formulation. Each formulation was analyzed in the range of dilutions, not exceeding their cytotoxic concentrations, as determined previously [18]. A significant decrease in the IL-6 levels measured in the cell culture medium was observed for six formulations: NG, B, S, NG:S-5, NG:C-7, and BG (Figure 8). The observed effect was concentration dependent to an extent and reached a 25–30% reduction in IL-6 production at 10 µM UA (NG) and 78 nM UA (S). In addition, a more significant decrease in IL-6 production was observed for NG:S-5 (at 10 µM UA) and NG:C-7 (at 5 µM UA), reaching 40–45%; however, the effect started at much lower concentrations of each formulation (even below 156 nM UA) and concentration dependency was limited. Bi-gel formulations (BG, Figure 8H) appeared to be the most effective in mitigation of IL-6 production. Results show a significant dampening of IL-6 production, resulting in the observed levels of this cytokine lowering to 50% of the M5-stimulated levels for BG at a 0.0625 µM concentration of ursolic acid. Further increases in the concentration induced even stronger reductions in IL-6, reaching 10% of the M5-stimulated levels. This effect was concentration dependent and was observed at lower UA concentrations than for ursolic acid alone. It needs to be noted that, although incubation of HaCaT cells with UA-containing balm (formulation B) resulted in the marked decrease in IL-6 production at low concentrations (312 nM UA), at higher UA concentrations, thus effect was not visible and it turned into stimulation of IL-6 production at concentrations higher than 2.5 µM. Similarly, the limiting effect of the S formulation of UA was weaker at concentrations higher than 78 nM UA, yet it did not result in the stimulation of IL-6 release, even at the highest UA concentration tested (2.5 µM UA).

Similarly, we analyzed IL-8 production in the same experimental configuration (Figure 9). Again, M5 stimulation increased production of IL-8, as observed previously. To our surprise, no decrease in IL-8 production could be observed for any of the tested formulations, except that of bi-gel (BG). Similarly to the reduction in IL-6, bi-gel decreased production of IL-8 by 50% at a 0.0625 µM UA concentration. Again, the reduction in IL-8 production was concentration dependent. At a 1 µM UA concentration in BG, IL-8 levels corresponded to ~20% of the maximal levels induced by M5. On the contrary, NG, S, NG:S-5, NG:C-7, and C all increased IL-8 production by M5-stimulated HaCaT cells. This effect was concentration dependent and most apparent for NEG formulation, where, even at a 156 nM UA concentration, the increase was significant. In the case of all other formulations, IL-8 levels were increased with higher concentrations of ursolic acid, reaching statistical significance only for the higher range of concentrations. B and NG:B-4 formulations had no effect on IL-8 levels in the HaCaT culture medium in comparison to the M5-stimulated control.

To verify the effectiveness of the prepared formulations in the inhibition of inflammatory-related keratinocyte proliferation, a model based on proliferation of M5-stimulated HaCaT cells was used again. As expected, stimulation with the psoriasis-related M5 cytokine mix induced a significant increase in HaCaT proliferation, observed 72 h after M5 stimulation. A concentration range of all formulations was tested; these formulations were kept below cytotoxic levels, as established previously. With the exceptions of NG:C-7 and B, all formulations showed a tendency to limit proliferation of unstimulated HaCaT cells (Figure 10). However, only S, NG:S-5, C, and BG were able to limit the proliferation of the M5-stimulated HaCaT cells to the levels of the unstimulated control. Bi-gel was the most effective, as a 0.3125 µM concentration of ursolic acid in bi-gel formulation significantly decreased cell proliferation, and further increases in UA concentrations up to 1.25 µM resulted in an increasingly prominent anti-proliferative effect. S and NG:S-5 were equally effective, as a 625 nM UA concentration in these formulations was enough to limit proliferation of M5-stimulated cells to the level of the unstimulated control, while C was effective at a 1.25 µM UA concentration. It is worth highlighting that the NG:B-4 formulation also showed a tendency to limit proliferation of M5-stimulated HaCaT cells at a 625 nM ursolic acid concentration, but this effect was accompanied by a decrease in the proliferation of unstimulated cells and was not statistically significant.

The presented research aimed at the determination of the effects of ursolic acid on the psoriatic-like phenotype of HaCaT cells induced by M5 cytokine stimulation. The results indicate the potency of ursolic acid in the limitation of IL-6 and IL-8 production and normalization of the induced keratinocyte hyperproliferation. These effects were accompanied by the induction of anti-inflammatory, apoptotic cell death in HaCaT cells and human neutrophils. These beneficial effects were mostly retained in the prepared formulations of ursolic acid, where bi-gel was highly effective in the reduction in M5-stimulated IL-6 and IL-8 production and hyperproliferation control.

## 4. Discussion

In recent years, multiple psoriasis-like in vitro cell culture models have been developed. These attempts included stimulation of keratinocyte-like cells (HaCaT cell line) with LPS, imiquimod, INF-γ, or psoriasis-related cytokines and their mixtures. These include the model based on the concurrent stimulation of the HaCaT culture with IL-17A, IL-22, oncostatin-M, IL-1α, and TNF-α [7]. This M5 stimulation model mimics typical features of psoriasis, including pro-inflammatory response and keratinocyte hyperproliferation, accompanied by inhibition of keratinocyte differentiation and increased expression of psoriasis-related biomarkers [21]. This model was implemented and characterized an elevated inflammatory response, evaluated by an increase in IL-6 and IL-8 production, augmentation of S100A7, S100A9, CXCL1, and CCL20 expression, and an increase in the proliferation in HaCaT cells, consistent with the psoriasis-like phenotype.

Ursolic acid, a pentacyclic triterpenoid initially isolated from bearberry (*Arctostaphylos uva ursi*), is a widespread compound of plant wax. Anti-inflammatory and pro-apoptotic activities of UA have been previously described [22,23,24] and include downregulation of IL-17 in Th_17_ T-cells, the main psoriasis facilitators [25]. Herein, an M5 stimulation model based on the HaCaT cells was implemented to evaluate the effects of ursolic acid on the development of the psoriatic phenotype. The anti-inflammatory effect of ursolic acid in this cell model was characterized and observed as a significant, concentration-dependent decrease in the M5-stimulated IL-6 and IL-8 release to the culture medium. This effect was accompanied by the ursolic acid-induced reduction in the expression of S100A7, S100A8, and S100A9 psoriasis biomarkers, the increase in HaCaT apoptosis, and the normalization of the hyperproliferative phenotype stimulated by the M5 cytokine mix. To the best of our knowledge, this is the first analysis of the effect of ursolic acid on the cell phenotype in the highly pro-psoriatic environment induced by M5 stimulation.

Interleukin 6 has been previously implicated in the development of psoriasis. For example, serum IL-6 levels were raised in active psoriatic patients [26] and were decreased early in response to infliximab anti-TNFα therapy in psoriatic patients [27]. IL-6 was detected in psoriatic lesions and patients’ sera, and was also found to stimulate keratinocyte proliferation [28]. At the same time, psoriatic monocytes and keratinocytes were able to produce IL-6 in a positive feedback loop [29]. Strikingly, in recent reports, IL-6/IL6R-mediated STAT3 activation in keratinocytes, rather than in immune cells, was responsible for the development of the psoriatic phenotype in a mouse model [30], highlighting the localized production and activity of IL-6 as being fundamental in psoriasis development. Herein, we demonstrate that ursolic acid displays the potential to decrease HaCaT-mediated IL-6 production in vitro in a highly psoriasis-stimulating environment. Because the response of T_reg_ cells to IL-6 dampens their immune suppression and may lead to the overstimulation of Th_17_ cells observed in psoriatic patients [31], this IL-6-reducing effect of ursolic acid may translate into the reduced activation of Th_17_ cells locally and lead to the normalized phenotype.

Even more notably, production of active IL-8 in psoriatic scales has been one of the hallmarks of psoriasis for years, as early reports indicated the presence of extracellular IL-8 immunostaining in psoriatic plaques [32]. IL-8 was identified as the main T-cell and neutrophile chemoattractant factor in psoriatic scale extracts and its levels were increased ~150-fold in comparison to healthy skin [33]. Similarly, the presence of the IL-8 receptor was increased in psoriatic patients [34], and recent bioinformatic analysis indicated CXCL-8 (IL-8)-related genes were a novel gene hub, related to 22 subtypes of immune cells essential in psoriasis [35]. IL-8 is responsible for the neutrophil influx and activation in the psoriatic plaques, and stimulates keratinocyte proliferation and angiogenesis, adding to the psoriasis pathogenesis [36]. Herein, we report the concentration-dependent limitation of HaCaT IL-8 production by ursolic acid. This limitation of local production of IL-8 is likely to reduce the neutrophil influx in the skin, resulting in the lowered inflammatory reaction and limitation of the IL-17A positive feedback loop in psoriasis. Both IL-8 and IL-6 are abundant in psoriatic lesions [37], indicating their local importance in the plaque development.

Expression of several psoriatic biomarkers, including S100A7 and S100A9, has been identified and shown to be induced in the M5 stimulation model [21]. For example, S100A7 (psoriasin) is an antimicrobial peptide, which also limits keratinocyte differentiation and induces their proliferation, mainly through IL-6 induction [38,39]. S100A8 and S100A9 (calgranulin A and B) are heterodimeric components of calprotectin, a potent neutrophil chemoattractant and inflammatory state marker [40]. Their levels are increased in the psoriatic stratum corneum and correlate with patients’ PASI score [41]. Herein, moderate effects of ursolic acid on the expression of these psoriatic biomarkers were observed. These include a ~50% reduction in S100A7, S100A8, and S100A9 expression, no effects on CXCL1, and a ~2-fold induction in CCL20. These results further indicate an ursolic acid-dependent reduction in the inflammatory state; however, induction of CCL20 may also indicate the potential of UA to upregulate psoriasis-promoting pathways. CCL20 is implicated in the accumulation of CCR6^+^ Th_17_ cells in the psoriatic skin [42]. Nonetheless, Th_17_ cells are not the only subpopulation of immune cells expressing the CCR6 receptor, as CCL20 also shows a promigratory effect for T_reg_ cells [42], and stimulation of CCL20 by oncostatin-M was found to be dispensable for psoriasis development in the animal model [43]. The complicated environment, which includes beneficial effects of ursolic acid on neutrophil chemoattractants and suppression of IL-6 production, contrasted by stimulation of CCL20 expression, does not allow for a definitive conclusion regarding the outcome and analysis of the effect of ursolic acid on Th_17_ and T_reg_ cell function, and thus warrants further investigations. However, the predominantly anti-inflammatory character of the action of ursolic acid tempts us to cautiously speculate that the overall effect of ursolic acid may be beneficial, and hindered IL-6 production may normalize T_reg_-Th_17_ cell interaction. It is worth highlighting that the effect of ursolic acid limits the keratinocyte response to IL-17a present in the M5 cytokine mix; hence, the downstream effects of Th_17_ cells’ activity are already subjected to UA limitation.

Aberrant proliferation of keratinocytes is a hallmark of psoriasis [44]. Here, we observed an increase in HaCaT cell proliferation upon M5 stimulation, consistent with the psoriatic phenotype. Ursolic acid decreased HaCaT proliferation in a concentration-dependent manner, normalizing the proliferation rate of M5-stimulated cells to the unstimulated cells at a 5 µM concentration. At higher concentrations, however, we observed increasing cytotoxicity of UA under conditions of low confluency, dividing the HaCaT population. Using Annexin V/propidium iodide staining, we identified, with flow cytometry analysis, apoptosis as a main mechanism of HaCaT cell death, accompanied by low levels of necrotic cells. Apoptosis is long-regarded as a beneficial mechanism in the control of keratinocyte proliferation in psoriasis. For example, UVB treatment of psoriatic lesions induces keratinocyte apoptosis [45] and vitamin D3 derivatives target hyperproliferation of psoriatic keratinocytes [46]. Interestingly, apoptosis is one of the natural mechanisms of resolving the psoriatic changes, as the apoptosis rate is reduced in established psoriasis, but significantly increases during the regression phase, even compared to the normal skin [47]. At the same time, the apoptosis-promoting mechanisms are disturbed in psoriasis, as psoriatic skin is resistant to INF-γ [48], is characterized by lowered levels of caspase-9 [49], and displays an imbalanced expression of apoptosis regulatory molecules [50]. As reported here, the pro-apoptotic effects of ursolic acid are similar to previously proposed therapeutic strategies, including the use of anthralin [51], calcipotriol [52], arsenic compounds [53], and even anti-TNF-α biologics [54,55].

Neutrophils were indicated in the psoriasis pathology for years, and neutrophil infiltrates have been identified in pre-psoriatic skin, even before keratinocyte hyperproliferation [56]. Local production of chemoattractive factors, rather than systemic stimulation of polymorphonuclear cells, was attributed to the neutrophil migration to psoriatic tissue [57]. Indeed, the analysis of the components of Munro micro-abscesses revealed neutrophils as the main cellular fraction, with surrounding keratinocytes expressing increased levels of IL-6 and IL-8 [58]. Inflammatory conditions promote neutrophil survival and limit their natural apoptosis rate, further exacerbating the inflammation state [59]. Rather, pro-inflammatory cell death—NETosis—is observed in psoriasis, which further enhances the inflammatory response of keratinocytes [60] and may lead to the release of psoriasis-related antigens and dendritic cell activators [61,62]. Recently, atypical neutrophil NETs were indicated as a source of LL-37/RNA complexes, which activate plasmoid dendritic cells and propagate psoriatic inflammation [63]. We therefore speculate that induction of apoptotic, non-inflammatory neutrophil cell death by ursolic acid may lead to the normalization of the PMN phenotype in psoriasis and shift the balance from the NET-dominated to the apoptosis-dominated neutrophil terminal fate. Pro-inflammatory stimulation provided by the mixture of oncostatin-M, IL-1α, TNF-α, IL-17, and IL-22 is expected to prolong neutrophil survival in psoriatic conditions. Indeed, the presented flow cytometry analysis hints at this effect, as M5-stimulated PMNs were characterized by a tendency to increase the number of alive cells, with 10% upregulation after 24 h. At the same time, under these psoriasis-mimicking, pro-PMN survival conditions, ursolic acid remained effective and induced high levels of apoptosis in a concentration-dependent manner, indicating the potential of UA to break through antiapoptotic neutrophil stimulation.

A moderate induction of reactive oxygen species in neutrophils was observed upon ursolic acid stimulation. ROS are believed to be important components of antibacterial activity of PMN and, when released, may add to the inflammatory reaction and tissue damage [64]. Contrasting reports on the role of ROS in psoriasis exist, possibly due to the induction of NETosis by high levels of reactive oxygen species; however, the beneficial role of ROS in psoriasis has recently been described. For example, imiquimod-induced psoriatic dermatitis is resolved by an increase in ROS, an effect attributed to the ROS-mediated hyperstimulation of T_reg_ cells [65]. Additionally, psoriatic inflammation induced by IL-17 is limited by reactive oxygen species in mice in vivo [66], and no change in ROS production was observed in IL-17 inhibitor, secukinumab-treated psoriatic arthritis patients, despite the positive response to therapy and resolution of symptoms [67]. This indicates that limitation of ROS production is not expected during the psoriatic treatment, and the local increase in ROS may be beneficial, at least when NETs are not induced effectively. Therefore, an ursolic acid-induced, moderate increase in ROS production, together with induction of non-inflammatory apoptosis, may further add to the beneficial effects of ursolic acid in the resolution of psoriasis symptoms.

Despite these beneficial effects of ursolic acid, the application of UA in topical therapy has remained rather limited, owning predominantly to the highly hydrophobic characteristic of the molecule. The effectiveness of the active ingredient largely depends on the form of its administration, which may reduce its cytotoxicity, increase penetration to the skin, and improve biological activity. A well-designed base formulation significantly influences the efficiency of cosmetic or pharmaceutical products. Both the physicochemical form of the vehicle and its composition may either increase or decrease the activity of the active substance [68,69]. Currently, a variety of UA formulations, such as micelles, liposomes, polymeric nanoparticles, and nanostructured lipid carriers, which can increase the solubility and bioactivity of UA, are described in the literature [70]. However, most of them are designed to be administered intravenously or orally and dedicated to use in various types of cancer treatment [71,72]. The anti-inflammatory activity of ursolic acid in topical formulations such as lipid-surfactant-based systems was described previously [17], as inhibition of edema induced by croton oil was observed. However, to the best of our knowledge, the biological effectiveness of topical formulations of ursolic acid is described herein for the first time.

Here, the biological effectiveness of ursolic acid formulations in psoriasis treatment was demonstrated, observed by the diminished IL-6 and IL-8 production and limiting effect on the M5-induced proliferation. The hybrid systems, such as nano-emulgel–serum (NG:S-5; with tamanu seed oil and sweet almond oil as specific emollients) and nano-emulgel–oleogel (BG; sunflower oil), were the most effective when the anti-inflammatory and anti-proliferative activities of ursolic acid were considered. Tamanu seed oil is characterized by a wide spectrum of biological activities, such as anti-oxidant, anti-inflammatory, and antibacterial properties and promotion of wound healing. Tamanu oil has been recommended for various skin issues, such as eczema, acne, psoriasis, burns, skin cracks, and dermatoses. Its resinous part is known to contain bioactive secondary metabolites mostly constituted by neoflavonoids such as calophyllolide [73,74].

Sunflower oil is composed of 55–70% linoleic acid and acts as an agonist at peroxisome proliferator-activated receptor-alpha (PPAR-α), which enhances keratinocyte proliferation and lipid synthesis. This essential fatty acid, together with the activation of the PPAR-alpha receptor, helps maintain the skin barrier and decrease trans-epidermal water loss [75]. It has been previously demonstrated that a decrease in PPAR-alpha levels was observed in the lesional skin of patients with psoriasis [76] and that topical PPAR-alpha agonist application decreased TNF-α and IL-1α in the skin [77]. Therefore, these emollients may have anti-psoriatic activities themselves and add to the anti-inflammatory effects of ursolic acid in these topical preparations. Combining nano-emulsion gel with macroemulsions containing a high proportion of liquid lipids, semisolids, and solid lipids in their formula allowed the acquisition of topical formulations characterized by soothing, moisturizing, and nourishing properties [18].

The presented study focused on in vitro models, which poses a significant limitation. The animal studies and human cohort analysis could indeed improve the importance of the findings and provide further validation. Such studies are planned as a follow-up analysis, as required during the registration process of the newly developed formulation described herein. Nonetheless, the anti-inflammatory properties of ursolic acid preparations in vivo were confirmed recently, as exemplified by protection from acute kidney injury in a mouse model [78], and its pharmacokinetic properties were reviewed previously [24].

To sum up, an in vitro model of psoriatic-like, pro-inflammatory stimulation using the mixture of IL-17A, IL-22, oncostatin-M, IL-1α, and TNF-α (M5 mix) was validated. The induction of the psoriatic-like phenotype in the cultured HaCaT cells was observed upon the stimulation, characterized by the increase in IL-6 and IL-8 production, expression of psoriatic biomarkers (S100A7, S100A9, CXCL1, and CCL20), and stimulation of keratinocyte-like cell proliferation. Furthermore, this model was used to test the effectiveness of ursolic acid as the inflammatory-limiting compound in this psoriasis-mimicking in vitro model. UA was identified as a potent anti-inflammatory agent, resulting in the inhibition of IL-6 and IL-8 production. The presented research demonstrated the UA-dependent hindrance of S100A7, S100A8, and S100A9 gene expression and UA anti-proliferation activity, based on non-inflammatory apoptosis induction in HaCaT cells. Further, a similar, proapoptotic effect on human neutrophil cells was confirmed, accompanied by moderate induction of ROS production and no pro-inflammatory stimulation of long-lived macrophages and skin fibroblasts in low to moderate concentrations. Finally, the biological effectiveness of ursolic acid-containing formulations was demonstrated, observed by the diminished IL-6 and IL-8 production and limiting effect on the M5-induced proliferation. Taken together, these results indicate ursolic acid may be a potential anti-psoriatic and anti-inflammatory agent with a promising bio-tolerance profile, and demonstrate its biological effectiveness in an appropriately designed vehicle formulation. Hybrid formulations, including bi-gel, were proven to be the most effective and exerted anti-inflammatory and anti-proliferative activities in the M5-stimulated HaCaT model. These formulations may be regarded as optimal preparations for topical ursolic acid delivery.

## Data Availability

The authors confirm that the data supporting the findings of this study are available within the article.

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
