# Peer review of "Ursolic Acid Formulations Effectively Induce Apoptosis and Limit Inflammation in the Psoriasis Models In Vitro"

_biomedicines, 2024, doi:10.3390/biomedicines12040732_

Round 1
Reviewer 1 Report
Comments and Suggestions for Authors
Comments to the editor
In this manuscript, the authors present of ursolic acid as potential anti-psoriatic and anti-inflammatory agent with promising bio-tolerance profile. Hybrid formulations, including bigel were proven to be the most effective and exerted anti-inflammatory and antiproliferative activities in the M5-stimulated HaCaT model.
To my opinion this paper was well written and bibliographical data are clearly presented. Therefore, it can be considered for publication after careful corrections. Detailed suggestions for revision are provided in the comments to the authors.
Comments to the authors
I have carefully read and evaluated the manuscript to be published in the journal “Biomedicines”. I considered that the manuscript should be modifications before published in the journal. My comments to improve their listed below:
1. What is the source of UA for this study? What are its specific benefits compared to other plant extracts?
2. Please check where the abbreviation and full name of the manuscript appear. For example, UA, h, etc.
3. In Figure 7B, is the treatment of the 1uM UA group inconsistent with that of the other UA concentration groups?
4. Whether to consider in vivo validation using animal models?
5. The quality of the figures in the manuscript needs to be further improved.
6. Why two cell lines, HaCaT and HSF, were chosen for in vitro experiments?
7. The manuscript mentions "Peripheral blood from human donors", was the ethical review approved?
Comments on the Quality of English Language
Comments to the editor
In this manuscript, the authors present of ursolic acid as potential anti-psoriatic and anti-inflammatory agent with promising bio-tolerance profile. Hybrid formulations, including bigel were proven to be the most effective and exerted anti-inflammatory and antiproliferative activities in the M5-stimulated HaCaT model.
To my opinion this paper was well written and bibliographical data are clearly presented. Therefore, it can be considered for publication after careful corrections. Detailed suggestions for revision are provided in the comments to the authors.
Comments to the authors
I have carefully read and evaluated the manuscript to be published in the journal “Biomedicines”. I considered that the manuscript should be modifications before published in the journal. My comments to improve their listed below:
1. What is the source of UA for this study? What are its specific benefits compared to other plant extracts?
2. Please check where the abbreviation and full name of the manuscript appear. For example, UA, h, etc.
3. In Figure 7B, is the treatment of the 1uM UA group inconsistent with that of the other UA concentration groups?
4. Whether to consider in vivo validation using animal models?
5. The quality of the figures in the manuscript needs to be further improved.
6. Why two cell lines, HaCaT and HSF, were chosen for in vitro experiments?
7. The manuscript mentions "Peripheral blood from human donors", was the ethical review approved?
Reviewer 2 Report
Comments and Suggestions for Authors
After a thorough review of the manuscript titled "Ursolic acid formulations effectively induce apoptosis and limit inflammation in psoriasis models in vitro," several key points and suggestions for improvement emerge:
1. The prerequisite for using the ΔΔCT method to calculate relative expression levels is that the amplification efficiencies of the target and reference genes must be similar or close to 100% to ensure the accuracy of the quantification. Please provide the corresponding amplification efficiency data.
2. For the manuscript, it is crucial to provide comprehensive details of the qPCR methodology employed, as per the requirements of the MIQE guidelines proposed by Stephen (doi: 10.1373/clinchem.2008.112797). Please provide the corresponding information.
3. The manuscript has collected human peripheral blood samples; is ethical approval required?
4. The authors should supplement the discussion section with a discussion on the limitations of this experiment, such as the fact that only in vitro experiments were conducted.
3
Reviewer 3 Report
Comments and Suggestions for Authors
This manuscript presented a complementary study to their previous study on ursolic acid different formulations. In this study the efficiency of ursolic acid alone and with formulations in limiting the inflammation in the psoriasis models (HaCaT cell model, stimulated with the M5 mixture of pro-psoriatic cytokines). The study is interesting, and the introduction and discussion are well-written. The references were new, and the conclusion was good. A few comments need to be revised as follows:
1. In the introduction section, the authors must clarify the previous study of ursolic formulation as it is not clearly stated.
2. In the methodology section:
- In line number 184, “as described here [76].” It must be written “as described by Bryzek et al., 2019 [76].
3. In results section:
- In figure 1. The concentrations of ursolic acid must be written 2.5, not 2,5. Also, figure 4 E needs revision of the apoptotic and necrotic percentages and other figures in the manuscript.
Reviewer 4 Report
Comments and Suggestions for Authors
Using in vitro model, the authors studied the effects of Ursolic acid (UA), on psora-sis-associated inflammation using skin and immune cells. UA mitigates inflammatory responses and hyperproliferation of keratinocytes, as well as a lack of negative influence on other cell types. The authors designed hybrid systems to enhance the bioavailability of UA. Expanding upon these discoveries, new topical formulations were created by combining UA with hybrid nanoemulsion and bigel topical systems to improve the absorption into the body and enhance its effectiveness.
Particularly, one of them, bigel, has shown significant efficacy in reducing the pathogenic reaction of keratinocytes to pro-psoriatic stimulation, surpassing the effectiveness of ursolic acid alone.
Altogether, the findings suggest that topical formulations containing ursolic acid demonstrate a favorable anti-inflammatory effect in laboratory settings and could potentially be used for treating psoriasis on the skin surface.
This is an interesting study. There are a few minor points.
1. The abstract should be summarized with findings.
2. Annexin histograms are not readable.
3. Figure 8 should be presented with high resolution or broken into two figures for better readability.
Comments on the Quality of English Languageappropriate
Round 2
Reviewer 1 Report
Comments and Suggestions for Authors
It can be pulished now
Comments on the Quality of English LanguageIt can be pulished now